# Constant Potential Coulometric Measurements with Ca^2+^-Selective Electrode: Analysis Using Calibration Plot vs. Analysis Using the Charge Curve Fitting

**DOI:** 10.3390/s22031145

**Published:** 2022-02-02

**Authors:** Anna Bondar, Konstantin Mikhelson

**Affiliations:** Chemistry Institute c/o, St. Petersburg State University, 26 Universitetsky Prospect, Stary Peterhof, 198504 St. Petersburg, Russia; amiyami67@mail.ru

**Keywords:** calcium ion activity, constant potential coulometry, analysis, calibration plot, charge curve fitting

## Abstract

The possibility of analysis using charge curve fitting in constant potential coulometric mode instead of using a calibration plot is explored, for the first time. The results are compared with the analysis based on the use of a calibration plot. A Ca^2+^ ion-selective electrode, with and without an electronic capacitor in series, is used as a model system in pure solutions of CaCl_2_. Both techniques delivered good results (error within 2%) when the final and the initial concentration values differed by not more than three times. Larger differences result in 10–25% error. The presence of an electronic capacitor in the measurement circuit and in series with the electrode, allows for significantly faster response.

## 1. Introduction

The constant potential coulometry method invented in the Bobacka’s group opened new opportunities for the use of ion-selective electrodes (ISEs) [1,2,3]. Originally, the features of the method were studied with K^+^-selective ISEs as a model system [1,2,3,4]. Further studies showed that this method can be used for quantification of divalent cations Ca^2+^, Pb^2+^, and Cu^2+^ [5,6,7], and NO_3_^−^, SO_4_^2−^, and ClO_4_^−^ anions [8,9]. The main advantage of the constant potential coulometry when compared with potentiometry is its ability to register small changes of target ion concentration. The sensitivity of the traditional potentiometric measurements with ISEs is limited by the Nernst factor: RT/z_I_F, where R, T, and F stand for the gas constant, temperature, and the Faraday constant, while z_I_ is the ion charge number. The constant potential coulometry method allows for overcoming this limitation. It was reported on registration of small changes of ion concentration, below 1%, which is especially attractive for practical applications [7,10]. An important modification of the approach was proposed by Bakker’s group: use of an electronic capacitor in series with the electrode [11,12,13]. In this way, it is possible to decrease the measurement time significantly without a loss in the sensitivity of the measurements, e.g., the pH in sea water was measured with the sensitivity of 0.001 pH units [12].

From the practical point of view, constant potential coulometry distinctively differs from zero-current potentiometry. In potentiometry, one can obtain a calibration plot and then use this plot once the ISE is immersed into the sample. This is because a potentiometric signal, the electromotive force (EMF) of a cell containing an ISE and a suitable reference electrode, arises spontaneously. Respectively, the EMF value registered in the sample can be directly translated into the sought ion concentration (rigorously, the activity) value using the calibration plot. If the calibration plot is stable over time, ideally, one can measure only in samples, always using the same calibration plot for analysis.

In the constant potential coulometric method, the recorded signal, current, arises upon a change of the concentration of ion in solution, while the ISE open circuit potential (OCP) is maintained constant by an instrument. The OCP is measured in an initial solution; when the same potential is applied to the electrode brought in contact with another (a final) solution, the current is then measured. The cumulated charge is obtained by integration of the current response over time. Respectively, both current and charge depend on the compositions of both solutions: the final and the initial. Thus, the calibration plot refers not to concentrations (rigorously, the activities) per se but to their changes. To plot a calibration curve, one carries out a series of dilutions of an initial standard solution or makes a series of additions to this standard. In this way a set of calibration solutions can be obtained. The measured charge values plotted vs. the activity of the target ion in the solutions form the calibration curve. Next, the ISE is immersed into one of the calibration solutions; the potential is recorded and then maintained constant while the solution is replaced with the sample. The measured charge shows the ratio of the activity of the ion in the calibration solution and in the sample. In other words, analysis with ISEs under constant potential coulometry mode, even if the calibration plot is absolutely stable, requires measurements in two solutions: the sample and one of the calibration solutions. The calibration solution chosen can be called the reference solution. Once the charge calibration plot is obtained it can also be used for analysis with standard additions/dilutions.

It is tempting, however, to try another approach to analysis with ISEs in the constant potential coulometric mode: to fit the current and the charge curves recorded to suitable equations [14,15], and then use the fitted parameters for the analysis. In this case, ideally, there is no need to obtain a calibration plot. Only two measurements must be done: in a reference solution and in the sample, and then the ratio of the two activities can be calculated.

For the first time, this option is explored in this paper. Among other ISEs, Ca^2+^ ISE (and K^+^ ISE) is the best studied [16], and therefore the most convenient for exploring a new technique of analysis. On the other hand, in many real samples, e.g., in blood and serum, only a part of calcium is present as free Ca^2+^ ions while the rest forms complexes with anions and proteins [17]. Therefore, to avoid complications, we used a Ca^2+^ ISE in pure solutions of CaCl_2_ as a model system. The results obtained are compared with those obtained via a calibration plot.

## 2. Materials and Methods

### 2.1. Chemicals and Materials

Calcium ionophore I diethyl N,N’-[(4R,5R)-4,5-dimethyl-1,8-dioxo-3,6-dioxaoctamethylene] bis(12-methylaminododecanoate) (ETH 1001), cation-exchanger sodium tetrakis[3,5-bis(trifluoromethyl)phenyl]borate (NaTFPB), lipophilic electrolyte tetradodecyl ammonium tetrakis(p-Cl-phenyl)borate (ETH 500), plasticizer 2-nitrophenyloctyl ether (oNPOE) were from Merck (Darmstadt, Germany), Selectophore grade. Ethylenedioxythiophene (EDOT) was from Fluorochem (Derbyshire, UK) and sodium polystyrene sulfonate (NaPSS) was from Aldrich (St. Louis, MO, USA), analytical grade. High molecular weight poly(vinyl chloride) (PVC), analytical grade, was from Ohtalen (St. Petersburg, Russia). Tetrahydrofuran (THF) was from Vekton, and distilled before use. CaCl_2_ (analytical grade) was from Reaktiv (Moscow, Russia). Aqueous solutions were prepared with deionized (DI) water with resistivity of 18.2 MΩ∙cm (Milli-Q Reference, Millipore, Burlington, MA, USA). The membrane cocktail contained PVC (360 mg), oNPOE (720 mg), ETH 1001 (9.9 mg), NaTFPB (6.4 mg), ETH 500 (16.5 mg) in 6.1 mL of THF. The mixture was gently mixed for 30 min using a Selecta Movil Rod (Barcelona, Spain) roller-mixer until clear solution was obtained.

### 2.2. Electrode Preparation

Glassy carbon electrodes representing glassy carbon rods (GC) with diameter of 3 mm, in Teflon bodies with outer diameter of 7 mm were from Volta (St. Petersburg, Russia). Adhesion of PVC membrane to Teflon is poor. Therefore, to prevent delamination of the membranes from the electrode bodies, prior to deposition of the conducting polymer (CP) layer, the electrodes were incapsulated in nonplasticized PVC tubes, as described elsewhere [15]. After that, the surfaces of glassy carbon rods were thoroughly polished on chamois leather with diamond slurry P/N 250.1030 on P/N 259.1025 substrate from Antec Scientific (Zoeterwoude, The Netherlands) and then with 0.3 mm alumina paste from Buehler (Lake Bluff, IL, USA). Then the electrodes were rinsed with DI water, placed into 1 M HNO_3_ for 5 min, rinsed with DI water, and sonicated in ethanol for 5 min (Elmasonic L15H, Elma, Wetzikon, Switzerland). Finally, the electrodes were sonicated for another 5 min in DI water.

The polyethylenedioxythiophene (PEDOT) layer on the surface of GC electrodes was formed by galvanostatic electropolymerization from solution containing 0.01 M EDOT and 0.1 M NaPSS in mixed aqueous-acetonitrile (9:1 *v*/*v*) solvent, as described elsewhere [7]. The estimated polymerization charge was 2.4 mC, and the estimated film thickness was 0.5 µm.

The membranes were formed by drop-casting 70 μL of the membrane cocktail on the top of the electrode, in two equal consecutive drops. The cocktail covered the whole surface: PEDOT + Teflon + PVC coating. The direct contact of the membrane layer with the PVC outer coating helped to prevent delamination of the membranes from the electrodes. The thickness of the membranes estimated from the dry mass of the cocktail and the electrode diameter was ca. 135 μm. Three replicate electrodes (electrodes 1, 2, 3) were used in the study.

### 2.3. Measurements

Zero-current potentiometric measurements were performed with an Ecotest 120 8-channel potentiometric station (Econix, Moscow, Russia). Electropolymerization procedure and nonzero current measurements were carried out with an Autolab 302N potentiostat–galvanostat (Metrohm, Herisau, Switzerland). The reference electrode in all measurements was Ag/AgCl in saturated KCl, with a custom-made flexible low-leak salt bridge filled with the same solution. The counter electrode in the nonzero current measurements was bare glassy carbon rod. Part of the measurements was performed with an electronic capacitor with a nominal value of 10 µF in series with the ISE. The quality control of the electrodes was performed in the zero-current potentiometric mode by sequential tenfold dilution of 0.1 M CaCl_2_ with a 700 Dosino / 711 Liquino system (Metrohm, Herisau, Switzerland). Chronoamperometric/coulometric measurements were performed with a time resolution of 0.03 s, by additions of suitable aliquots of CaCl_2_ to the initial solution 0.25 mM aqueous CaCl_2_. For the improvement of the signal-to-noise ratio, the procedure of the chronoamperometric measurements was performed as described elsewhere [7]. The measurements were carried out at 20 ± 1 °C.

## 3. Results

### 3.1. Control of the Quality of the ISEs

Prior to chronoamperometric/coulometric measurements, the ISE response to Ca^2+^ in pure CaCl_2_ and in model solutions was controlled by the EMF measurements. Calcium ion activity coefficients in pure and in mixed solutions were calculated by the Davies equation (the third approximation of the Debye–Hückel theory), using 6 as the Kielland parameter for Ca^2+^ [16]. In each solution, the EMF was recorded for 300 s, and the average signal for the last 100 s was used to plot the calibration curves presented in Figure 1. The SDs never exceeded 1 mV and therefore are not shown. The standard EMF values vary between the three replicate electrodes, which is typical for solid-contact ISEs [16]. The theoretical value of the Nernstian slope for a divalent cation is 29.07 mV/log(*a_C_*_a_) at 20 °C. One can see that the ISEs showed practically ideal Nernstian response over the concentration range of CaCl_2_ from 0.1 to 10^−6^ M.

### 3.2. Chronoamperometric/Coulometric Measurements

#### 3.2.1. Electrodes without Electronic Capacitor in Series

Pure solutions of CaCl_2_ with concentrations 0.25, 0.5, 1, 2, 4, and 8 mM were used as calibration standards. Coulometric calibration plots were obtained as follows. Starting with 0.25 mM CaCl_2_, sequential two-fold additions were made, obtaining solutions with 0.5, 1, 2, 4, and 8 mM CaCl_2_. At each step, the OCP was recorded in the initial solution, and then current was recorded upon the addition, for 100 s. As an example, the current response plot obtained with electrode 1 is shown in Figure 2a. Other electrodes showed analogous responses. Charge curves were obtained by integration of current over time. The charge response plot obtained with electrode 1 is shown in Figure 2b; other electrodes showed analogous behavior. The coulometric calibration plots obtained using charge values cumulated during 100 s are shown in Figure 3.

One can see that the coulometric calibration plot can be expressed in a way similar to that used for the EMF in potentiometry:(1)$$Q=Q^{0}+S_{q}log\left(a_{Ca}\right)$$
where *Q* is the cumulated charge. In analogy with the zero-current potentiometric measurements, *Q*^0^ can be called standard charge value, it refers to $a_{Ca}=1$, and *S_q_* is called the coulometric slope. Note that the value of *Q*^0^ for the same electrode is dependent on the initial concentration used in the measurement procedure. In this study, the latter was 0.25 mM CaCl_2_. The *Q*^0^ and *S_q_* values obtained by linear fitting of the data shown in Figure 4 for electrodes 1, 2, 3 are presented in Table 1.

The *Q*^0^ standard charge value and *S_q_* the coulometric slope depend on the resistance and on the capacitance of the individual electrodes. Once, for a particular electrode, values of *Q*^0^ and *S_q_* are known, the sought activity of the target ion in a sample can be obtained using Equation (1). The results are described in Section 4.

Another approach implies fitting the charge curves recorded in the sample solutions to suitable equations. A solid-contact ISE can be represented as *R_Mem_* resistor (primarily the resistance of ion-selective membrane) and *C_CP_* capacitor (primarily that of the CP layer) in series [14]. By a time *t* after an abrupt change of $a_{I}^{Ini}$—the initial activity value to $a_{I}^{Fin}$—the final value, *Q*_(*t*)_ charge is cumulated. Its value can be described as
(2)$$Q_{\left(t\right)}=\frac{RT}{z_{I}F}ln\frac{a_{I}^{Ini}}{a_{I}^{Fin}}\left[C_{CP}\left(1-e^{-\frac{t}{R_{Mem}C_{CP}}}\right)\right]$$
where *R*, *T*, and *F* are gas constant, temperature, and Faraday constant. A current flow across an electrode may result in a concentration polarization in the electrode membrane. It was shown that the possible impact from the concentration polarization in the membrane can be described by adding the Cottrellian term [15]:(3)$$Q_{\left(t\right)}=\frac{RT}{z_{I}F}ln\frac{a_{I}^{Ini}}{a_{I}^{Fin}}\left[C_{CP}\left(1-e^{-\frac{t}{R_{Mem}C_{CP}}}\right)+N\sqrt{t}\right]$$

Factor *N* is dependent on *A*_E_, the electrode surface area; *C_I_*, the concentration of the ionic species in the membrane; and *D_I_*, their diffusion coefficient: $N=2\pi^{-1/2}\left(F/RT\right)A_{E}C_{I}D_{I}$ [18]. Experimentally obtained charge curves were fitted to Equations (2) and (3). As a typical example, results referring to electrode 1 when 2 mM CaCl_2_ was replaced with 4 mM are presented in Figure 4.

With time resolution 0.03 s, experimental points in Figure 4, if all shown, are too close to one another and hinder the visualization of the results. Therefore, for better visibility, only part of the experimental points is shown. One can see that fitting the curve to Equation (3) with the Cottrellian term is significantly better than to Equation (2). This is consistent with the observations reported elsewhere [15]. On the other hand, although each individual curve can be nicely fitted to Equation (3), the fitted parameters *R_Mem_*, *C_CP_*, and *N* vary from one individual curve to another. In particular, for *R_Mem_* the membrane bulk resistance depends on the concentration of the solution and correlates with sorption of water by membranes [19,20,21,22,23,24]. Because of this variation, the use of the respective average values obtained in calibration solutions (0.25, 0.5, 1, 2, 4, 8 mM) results in scattering in the sample analysis data, as discussed in Section 4.

#### 3.2.2. Electrodes with Electronic Capacitor in Series

Experiments analogous to those described in Section 3.2.1 were carried out with the same electrodes, but with an electronic capacitor (10 µF) connected in series. The current curves recorded with electrode 1 are shown in Figure 5a. For comparison, the curve referring to the concentration change from 0.25 to 0.5 mM obtained with the same electrode without an electronic capacitor in series is also presented in the figure. The cumulated charge curves obtained by integration of the current over time are shown in Figure 5b. It can be seen clearly that the current and the charge responses are much faster. This is consistent with the capacitance values obtained by fitting the curves; 42.1 µF in the absence of electronic capacitor decreased to 5.86 µF when the capacitor was connected in series with the electrode. Two other replicate electrodes showed analogous results. These data are consistent with those reported by the Bakker’s group [11,12,13]. However, even with an electronic capacitor in series, the charge curves did not saturate even at 100 s. The reason for this is the impact from the Cottrellian term in the signal. Therefore, to explore the possibilities of faster calibration and analysis, we used the charge values cumulated by 15 s, in contrast to 100 s used for electrodes without the electronic capacitor. The coulometric calibration plots obtained using charge values cumulated during 15 s are shown in Figure 6.

The calibration plots obtained follow Equation (1). The *Q*^0^ values for electrodes 1, 2, 3 were, respectively, −1.227·10^−6^, −1.178·10^−6^, and −9.153·10^−6^. The slopes were −3.363·10^−7^, −3.217·10^−7^, and −2.483·10^−7^ C/log(a_Ca_). The results of the analysis of sample solutions with the aid of a calibration plot using Equation (1) are described in Section 4.

Similar to the case when there was no electronic capacitor, the charge curves were fitted to Equations (2) and (3). As a typical example, results referring to electrode 1 when 2 mM CaCl_2_ was replaced with 4 mM are presented in Figure 7.

For better visibility in Figure 7, as in Figure 4, only part of the experimental points is shown. One can see that fitting the curve to Equation (3) with the Cottrellian term is significantly better than to Equation (2). This is consistent with the results obtained in the absence of the electronic capacitor, and with those reported earlier [15]. Additionally, similar to the case when no electronic capacitor is present in the circuit, the fitted parameters *R_Mem_*, *C_CP_*, and *N* vary from one individual curve to another, resulting in scattering in the analysis data, as discussed in Section 4.

## 4. Discussion

Control of calcium is especially important in clinical chemistry, and the normal level of ionized calcium in blood and serum is ca. 1 mM [17]. Therefore, the sample solutions chosen covered the range around 1 mM: 0.4, 0.6, 0.8, 1.2, and 3.5 mM. These solutions were not used in calibrations. Concentration of Ca^2+^ in these solutions was measured by the constant potential coulometry method using calibration curves and individual curve fitting. The initial solution always was 0.25 mM CaCl_2_. The results presented in Table 2 and Table 3 refer to average data obtained with all three electrodes, without (Table 2) and with electronic capacitor in series (Table 3).

One can see that when the ratio of the initial and final concentration values is not large, e.g., final value is 0.8 mM while the initial is 0.25 mM, the use of a calibration plot delivers nice results. In these cases the error does not exceed 2% with and without an electronic capacitor in series with the ISE. Increase of the difference between the two solutions, initial and final, results in a drastic increase of the analysis error. Analysis using curve fitting delivers worse results than with the aid of the calibration plot. The main reason for this is variation of the fitted parameters from one individual curve to another, although each curve separately is nicely fitted to Equation (3).

Use of an electronic capacitor, although resulting in a much faster response as shown in Figure 6, did not ensure saturation of the charge curve even after 100 s (not shown). Obviously, this is because of a significant effect from the concentration polarization manifested by the Cottrellian term. Thus, in further studies, the membrane composition and thickness must be modified to minimize this effect.

## 5. Conclusions

The results of this study are in line with similar studies aimed at exploring the possibility of analysis under constant potential coulometry mode. This new method is especially promising when small changes of the analyte concentration must be registered [7,8,11,12,13]. It is therefore important that both techniques compared here delivered good results when the ratio of the final and the initial concentration values were relatively small. However, use of the entire calibration plot ensures better results. In further studies, it is worth modifying the electrode and the circuit to increase the precision of the analysis, whenever the analyte concentration changes are small or large. Obviously, both approaches to analysis—using calibration plot and using curve fitting—must be studied with different ISEs for different target analytes.

## Figures and Tables

**Figure 1 sensors-22-01145-f001:**
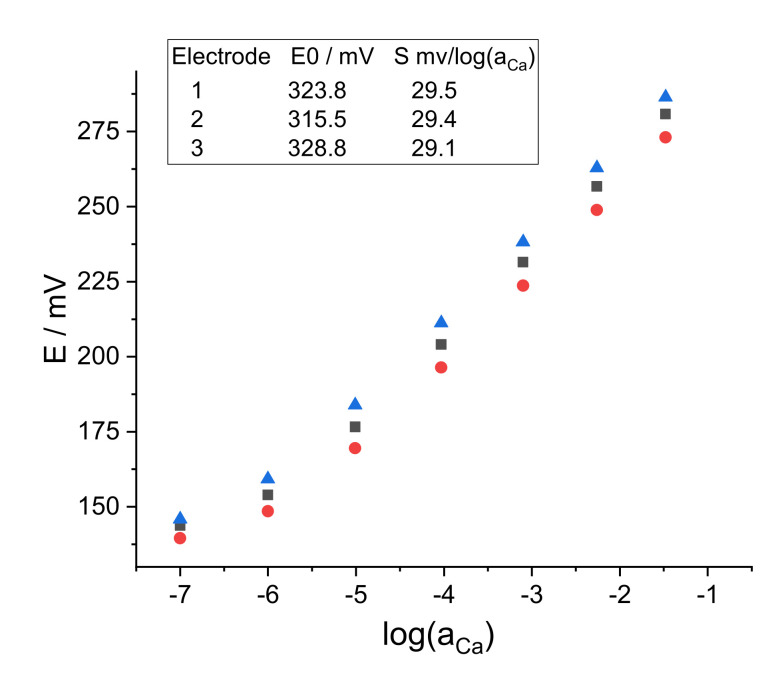
Potentiometric calibration of the ISEs. Square-1, circle-2, triangle-3.

**Figure 2 sensors-22-01145-f002:**
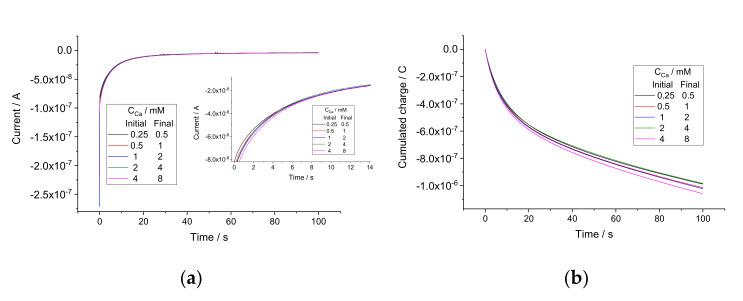
Current (**a**) and charge (**b**) responses of electrode 1 to sequential two-fold additions. Inset in (**a**) shows that the curves are similar but not the same.

**Figure 3 sensors-22-01145-f003:**
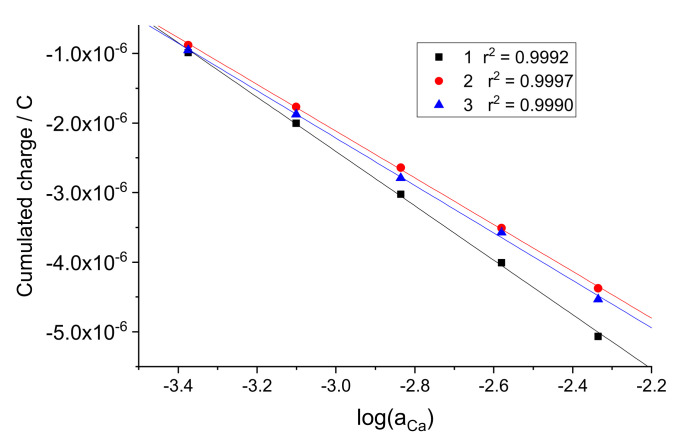
Charge calibration plots obtained with electrodes 1, 2, and 3. Data on log(a_Ca_) refer to activities of Ca^2+^ ion in solutions after sequential two−fold additions to the initial 0.25 mM CaCl_2_ with $log\left(a_{Ca}^{Ini}\right)=-3.655.$

**Figure 4 sensors-22-01145-f004:**
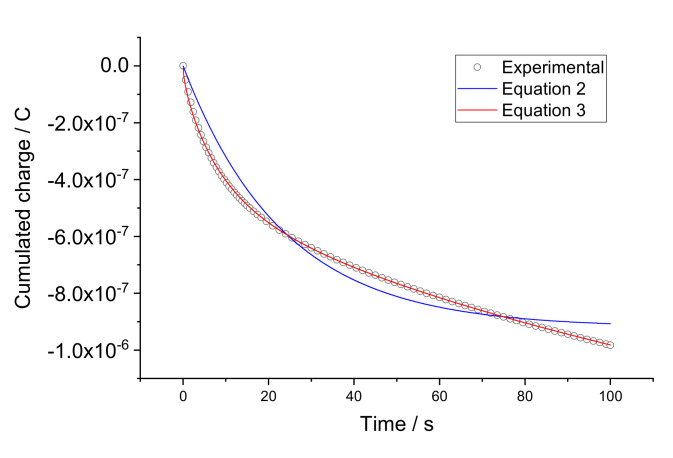
Fitting experimental charge curves to Equations (2) and (3). The experimental results were obtained with electrode 1 when 2 mM CaCl_2_ was replaced with 4 mM.

**Figure 5 sensors-22-01145-f005:**
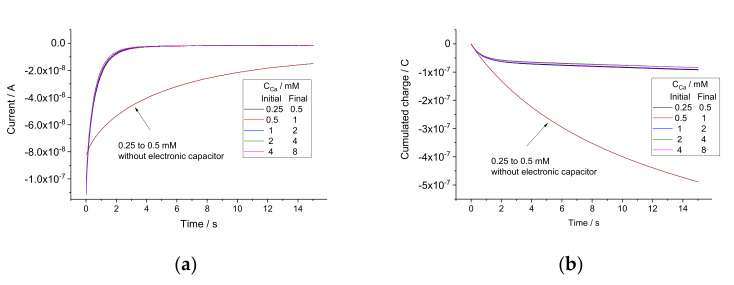
Current (**a**) and charge (**b**) responses of electrode 1 (in series with electronic capacitor) to sequential two-fold additions. Responses to replacement 0.25 mM with 0.5 mM CaCl_2_ in the absence of electronic capacitor are shown for comparison.

**Figure 6 sensors-22-01145-f006:**
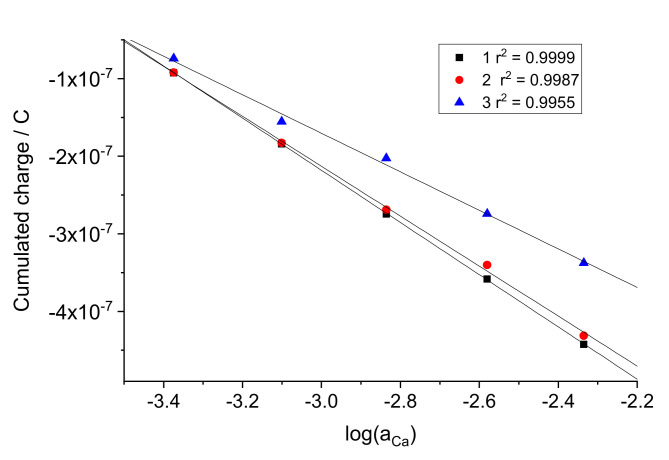
Charge calibration plots obtained with electrodes 1, 2, and 3 in series with electronic capacitor. Data on log(a_Ca_) refer to activities of Ca^2+^ ion in solutions after sequential two-fold additions to the initial 0.25 mM CaCl_2_ with $log\left(a_{Ca}^{Ini}\right)=-3.655.$

**Figure 7 sensors-22-01145-f007:**
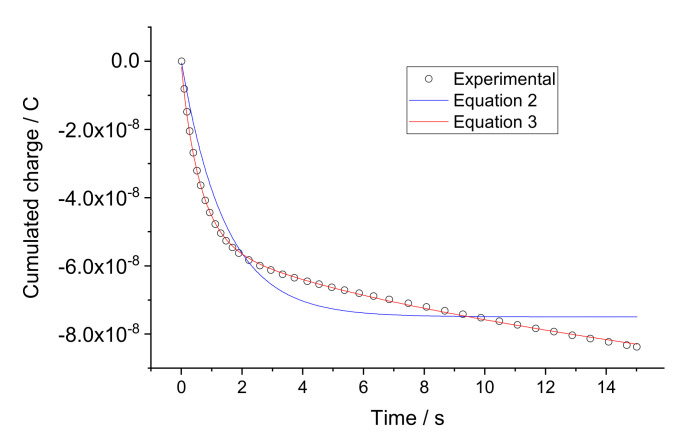
Fitting experimental charge curves to Equations (2) and (3), obtained with electrode 1 in series with electronic capacitor. The experimental results were obtained with electrode 1 when 2 mM CaCl_2_ was replaced with 4 mM.

**Table 1 sensors-22-01145-t001:** Coulometric calibration plot parameters of the ISEs.

	Electrode		
	1	2	3
	Without capacitor in series		
*Q*^0^ (C)	−1.413·10^−5^	−1.219·10^−5^	−1.244·10^−5^
*S_q_* (C/log(a_Ca_)	−3.907·10^−6^	−3.358·10^−6^	−3.408·10^−6^
	With electronic capacitor of 10 µF in series with the electrode		
*Q*^0^ (C)	−1.227·10^−6^	−1.178·10^−6^	−9.153·10^−6^
*S_q_* (C/log(a_Ca_)	−3.363·10^−7^	−3.217·10^−7^	−2.483·10^−7^

**Table 2 sensors-22-01145-t002:** Measurements of concentration of Ca^2+^ in sample without electronic capacitor.

	Using Calibration Plot			Using Curve Fitting		
Target, mM	Measured, mM	SD, mM	Recovery, %	Measured, mM	SD, mM	Recovery, %
0.400	0.403	0.004	101.8	0.387	0.003	96.7
0.600	0.612	0.003	102	0.589	0.004	98.1
0.800	0.802	0.013	100.2	0.870	0.006	109
1.200	1.15	0.05	95.5	1.17	0.08	97.7
3.500	4.14	0.06	118	4.38	0.20	125

**Table 3 sensors-22-01145-t003:** Measurements of concentration of Ca^2+^ in sample with electronic capacitor in series.

	Using Calibration Plot			Using Curve Fitting		
Target, mM	Measured, mM	SD, mM	Recovery, %	Measured, mM	SD, mM	Recovery, %
0.400	0.395	0.007	98.2	0.368		92.1
0.600	0.612	0.032	102	0.650	0.006	108
0.800	0.819	0.009	102	0.853		107
1.200	1.28	0.07	107	1.33	0.35	111
3.500	3.93	0.10	112	4.02	0.80	115

## Data Availability

The data presented in this study are available in this article.

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
