# Peer review of "Constant Potential Coulometric Measurements with Ca2+-Selective Electrode: Analysis Using Calibration Plot vs. Analysis Using the Charge Curve Fitting"

_sensors, 2022, doi:10.3390/s22031145_

Round 1

Reviewer 1 Report

Major remarks:

1. The Introduction is rather weak. It contains some basics on the potential coulometry. At the same time some deeper analysis and discussion is required for the modern papers on the topic. Especially it is essential to analyze what have been already done (solutions and achievements) to overcome the issues mentioned in the Inroduction by authors.

2. The practical value of the presented research is doubtful. The authors declare the possibility of the analysis in the constant potential coulometric mode using the charge curve fitting is explored, for the first time. At the same time, as shown in Table 1 and 2, the approach using callibration plot superseeds the proposed one in all the points of comparison.

3. There were only 3 concentration values used for sample with and without capacitor, moreover 3.5 mM was the same point. It is unclear how the points' quantity and the concentaration values were chosen.

4. I disagree with the statement "use of calibration plot delivers nice results: the error is below 1% without electronic capacitor in series, and 2% with electronic capacitor", since this comparison made for the different concentration values. It is unclear why authors do that.

5. The manuscript text require certain English proofreading. There are many sentences with incorrect structure, unnecessary additional words or uncommon spelling, which make difficulties to understand the main idea of the statement. Some sentences, especially in Section 3 Results are extrremely difficult to read. Please split them by few parts or modify accordingly.

Minor remarks:

6. It would be better to put the main descriptions of the chemicals and materials, electrodes from section 2 to specific tables with names/models, parameters and charactersitics of the used hardware and chemicals. It will simplify data perception by readers.

7. Please improve quality of Figure 2 (a) and (b), Figure 5 (a) and (b).

8. Please provide reference and/or clear justification for equations (1) and (2). If some parameters/constants were different from the source taken please explain. 

9. The values of Q0 and slope values in section 3 as well as some other would be better to show summing up as a table.

10. Please check Table 2 seems to lack 'Using curve fitting' heading.

Author Response

Replies to the Reviewers

We are very grateful to the Reviewers for careful reading and evaluation of our manuscript. We tried our best improving the manuscript in view of the comments and suggestions raised by the Reviewers. In view of the Reviewers’ comments, we carried our additional measurements, enlarged the Introduction and added several new references. In view of comments raised by Reviewers 2 and 3 we had to include additional references to our own studies. Below, we reply to the Reviewers’ comments one by one.

Reviewer 1

  1. The Introduction is rather weak. It contains some basics on the potential coulometry. At the same time some deeper analysis and discussion is required for the modern papers on the topic. Especially it is essential to analyze what have been already done (solutions and achievements) to overcome the issues mentioned in the Inroduction by authors.

We have extended the Introduction and added several relevant references.

  1. The practical value of the presented research is doubtful. The authors declare the possibility of the analysis in the constant potential coulometric mode using the charge curve fitting is explored, for the first time. At the same time, as shown in Table 1 and 2, the approach using callibration plot superseeds the proposed one in all the points of comparison.

We explored the possibility of the analysis with ISEs in the constant potential coulometry mode by the charge curve fitting approach. We do not claim that this approach delivers better results than the approach based on the calibration plot. For better clarity we modified the Conclusions accordingly.

  1. There were only 3 concentration values used for sample with and without capacitor, moreover 3.5 mM was the same point. It is unclear how the points' quantity and the concentaration values were chosen.

We carried out additional experiments, and now the data are provided for 5 concentration values. The sample sets are the same for ISEs with and without capacitor. Control of calcium is especially important in clinical chemistry, and the normal level of ionized calcium in blood and serum is ca. 1 mM. Therefore, the sample solutions covered the range around 1 mM: from 0.4 to 3.5 mM. The text is modified accordingly.

  1. I disagree with the statement "use of calibration plot delivers nice results: the error is below 1% without electronic capacitor in series, and 2% with electronic capacitor", since this comparison made for the different concentration values. It is unclear why authors do that.

Additional measurements showed that when the ratio of the initial and final concentration values is relatively small (e.g. 0.8/0.25) the error does not exceed 2%. The text is modified accordingly.

  1. The manuscript text requires certain English proofreading. There are many sentences with incorrect structure, unnecessary additional words or uncommon spelling, which make difficulties to understand the main idea of the statement. Some sentences, especially in Section 3 Results are extremely difficult to read. Please split them by few parts or modify accordingly.

We did our best improving the English of the paper. Also, we have split long sentences in parts.

Minor remarks:

  1. It would be better to put the main descriptions of the chemicals and materials, electrodes from section 2 to specific tables with names/models, parameters and charactersitics of the used hardware and chemicals. It will simplify data perception by readers.

We added the quality data on the chemicals used. The parameters and characteristics of the instruments are available in the respective Internet sites. We therefore believe adding such a table is hardly justified.

  1. Please improve quality of Figure 2 (a) and (b), Figure 5 (a) and (b).

Done. Lines are thicker, figures are larger. Figure 2 (a) now contains an inset for better clarity.

  1. Please provide reference and/or clear justification for equations (1) and (2). If some parameters/constants were different from the source taken please explain.

Equation (1) describes data shown in Fig. 3. Apparently, the Reviewer means equations (2) and (3). It is stated in the text that equation (2) describes the ISE as a resistor and a capacitor in series. This equation was derived elsewhere (ref. 9 in the original submission, ref. 14 in the revised text). Equation (3) also takes into consideration the charge transportation (diffusion/migration of ions) in the membrane. This equation was proposed elsewhere (ref. 10 in the original submission, ref. 15 in the revised text). The parameters result from fitting the experimental curves to these equations.

  1. The values of Q0 and slope values in section 3 as well as some other would be better to show summing up as a table.

Done, this is Table 1 in the revised manuscript.

  1. Please check Table 2 seems to lack 'Using curve fitting' heading.

Corrected.

Reviewer 2 Report

Dear Editor,

I have been asked to review the MS "Constant potential coulometric measurements..." Authors:  A. Bondar and K. Mikhelson.

As a summary: I think that the MS is suitable for the readers of sensors, especially for the special issue "Ion selective sensors and their applications".

The MS shows interesting experiments on the determination of low concentrations (activities) of Ca2+ with the coulometric measurement at constant potential using ion-selective electrodes. Two different methods are described in detail: The use of calibration curves and individual curve fitting. Fitting the experimental data (charge versus activity) shows a slight difference between the two equations with and without Cottrell term. However, two questions remain open:

What is the fitted value for the diffusion coefficient, and what does "electrode cross-sectional area" mean?

In addition, it is shown that the use of a capacitor with suitable capacity significantly reduces the measurement time, apparently without loss of sensitivity.

Furthermore, the authors show the disadvantage of the methods: If the concentration between sample and so-called reference solution is too large, both methods show larger deviations. Why?

I think the preparation of the membrane is quite tricky. How long is the preparation time and what is the stability of the homemade ion-selective electrode?

Last but not least: The authors claim that the sensitivity of the method used is much higher than that of conventional potentiometry. Can you quantify this?

All chapters, Introduction, Materials and Methods, Results and Discussions are suitable to introduce both experts but also newcomers in this field to the methods used.

After correcting these few monita, I recommend the MS for publication in Sensors.

Yours sincerely,

Reviewer

Author Response

Replies to the Reviewers

We are very grateful to the Reviewers for careful reading and evaluation of our manuscript. We tried our best improving the manuscript in view of the comments and suggestions raised by the Reviewers. In view of the Reviewers’ comments, we carried our additional measurements, enlarged the Introduction and added several new references. In view of comments raised by Reviewers 2 and 3 we had to include additional references to our own studies. Below, we reply to the Reviewers’ comments one by one.

Reviewer 2

  1. What is the fitted value for the diffusion coefficient, and what does "electrode cross-sectional area" mean? In addition, it is shown that the use of a capacitor with suitable capacity significantly reduces the measurement time, apparently without loss of sensitivity.

In the revised text the term "electrode cross-sectional area" is replaced with "electrode surface area". The diffusion coefficient was not fitted. The N parameter (fitted) contains a multiple of concentration and diffusion coefficients of ions in the membrane. Electrolytes in PVC membranes are only partly dissociated. Therefore, only the total concentration of the electrolyte is known by the membrane preparation, not that of ions. Other techniques: radio tracer measurements or chronopotentiometric measurements allow estimating the diffusion coefficients of ions in such membranes as 10−8  cm2/s.

  1. Furthermore, the authors show the disadvantage of the methods: If the concentration between sample and so-called reference solution is too large, both methods show larger deviations. Why?

In general, it is typical that errors accumulate along “distance” from the reference point. More specifically, in this case, additional error stems from the non-constancy of the membrane bulk resistance. Ideally, within the Nernstian potentiometric response range of an ISE, the bulk resistance must be constant. Our systematic studies over several recent years showed that this is not true: the membrane bulk resistance depends on the concentration of the solution, even in the absence of the co-extraction of the solution electrolyte. The effect of the variation of the resistance accumulates along the variation of the concentration. In the revised text, we briefly explain this. Because of this we had to add several references to our own works.

  1. I think the preparation of the membrane is quite tricky. How long is the preparation time and what is the stability of the homemade ion-selective electrode?

It may be tricky for a newcomer. However, our scientific group deals with ionophore-based ISEs since late 1970-s, so we have experience, instruments and chemicals. When everything needed is at disposal, it may take 2-3 h to make an ISE. The stability in terms of the lifetime of the homemade ISEs (as of any ISEs) is about several months.

  1. Last but not least: The authors claim that the sensitivity of the method used is much higher than that of conventional potentiometry. Can you quantify this?

This is not “our” claim, in the first place: we provided the respective references. Therefore, we did not modify the text in the revised manuscript. However, we are eager to explain here. The constant potential coulometry allows to register 0.1% change of the concentration of the analyte. This is equivalent to 25.6 µV in potentiometry. In principle, the modern instruments can register such small effects. However, in potentiometry, one cannot say whether the concentration did change, or this is just a drift. One can be sure about changes of 0.2 mV or larger. This translates into 0.8% change on the concentration. In the constant potential coulometry mode, very characteristic current peaks appear, justifying the interpretation, so 0.1% can be registered for certain.

  1. All chapters, Introduction, Materials and Methods, Results and Discussions are suitable to introduce both experts but also newcomers in this field to the methods used.

Frankly, we did not understand whether this is critics or approval.

Reviewer 3 Report

The authors report the analysis in the constant potential coulometric mode using charge curve fitting. The work is nicely presented and the result reported is of great interest to readers. However, some part of the manuscript is not well written. I would recommend the publication of the work if the following questions are properly resolved. 

  1. In the introduction part, the significance of the work is not clearly described. Analysis with ISEs under constant potential coulometry mode using calibration plot requires measurements in two solutions. While the analysis with ISEs by curve fitting requires two measurements. What are the advantages of the curve fitting method reported in this work?
  2. Besides, how does the work compare with the previous studies? Similar curve fitting equations have been adapted in Ref. 9 and 10. How does the work compare with these recent studies? What is the difference between the studies? And what is the unique contribution of the present work to the field?
  3. In the introduction part, only a limited number of references (12) are cited on the introduction. The background of the research, related works, and the rationale to conduct the research should be supported by the references. More references on the field are desired to give a brief introduction on the field for the readers that are not familiar with the field.
  4. The authors investigate the curve fitting method by the model system CaCl2. How is the generality of the model system? Does the conclusion apply to other solutions?
  5. In Figures 4 and 7, the experiment results are nicely fitted by Equ. 3. What is the physical foundation for the fitting equation? Is there any physical insights obtained from the fitting equation? What is the rationale for the variation of the fitted parameters Rmem, Ccp, and N?
  6. The caption of Fig 4 is not clear and does not describe the condition for the experimental curve.
  7. Some paragraphs have only one sentence, such as l132, l150, l227. Should the sentence be combined with the aforementioned paragraph or expanded with more discussion? 
  8. The table 1 and 2 are aligned to the left of the page. The layout of the table should be checked.

Author Response

Replies to the Reviewers

We are very grateful to the Reviewers for careful reading and evaluation of our manuscript. We tried our best improving the manuscript in view of the comments and suggestions raised by the Reviewers. In view of the Reviewers’ comments, we carried our additional measurements, enlarged the Introduction and added several new references. In view of comments raised by Reviewers 2 and 3 we had to include additional references to our own studies. Below, we reply to the Reviewers’ comments one by one.

Reviewer 3

  1. In the introduction part, the significance of the work is not clearly described. Analysis with ISEs under constant potential coulometry mode using calibration plot requires measurements in two solutions. While the analysis with ISEs by curve fitting requires two measurements. What are the advantages of the curve fitting method reported in this work?

Analysis with a calibration plot requires more than two measurements because the calibration plot must be obtained. Theoretically, this can be made only once (if the plot is ideally stable). However, it is not so, and calibration, typically, must be run on a daily basis. The text is modified for better clarity.

  1. Besides, how does the work compare with the previous studies? Similar curve fitting equations have been adapted in Ref. 9 and 10. How does the work compare with these recent studies? What is the difference between the studies? And what is the unique contribution of the present work to the field?

The authors of ref. 9 (now ref. 14) derived equation (2) but did not use it for analysis. In ref. 10 (now ref. 15) we modified the equation adding the Cottrellian term and obtained better fitting. However, the analysis was performed with the aid of a calibration plot. It is stated in the Introduction that analysis using curve fitting is explored here for the first time.

  1. In the introduction part, only a limited number of references (12) are cited on the introduction. The background of the research, related works, and the rationale to conduct the research should be supported by the references. More references on the field are desired to give a brief introduction on the field for the readers that are not familiar with the field.

The Introduction is enlarged, and several relevant references are added and discussed briefly.

  1. The authors investigate the curve fitting method by the model system CaCl2. How is the generality of the model system? Does the conclusion apply to other solutions?

The validity of the method depends on the performance of the respective ISE, rather than on the nature of the solution. Apparently, if an ISE in a given solution works well in the potentiometric mode, one can envisage promising results also in the constant potential coulometry mode. However, this must be studied experimentally. The text (Conclusions) is modified accordingly.

  1. In Figures 4 and 7, the experiment results are nicely fitted by Equ. 3. What is the physical foundation for the fitting equation? Is there any physical insights obtained from the fitting equation? What is the rationale for the variation of the fitted parameters RmemCcp, and N?

It is stated in the text that equation (2) describes the ISE as a resistor and a capacitor in series. This equation was derived elsewhere (ref. 9 in the original submission, ref. 14 in the revised text). Equation (3) takes into consideration also the charge transportation (diffusion/migration of ions) in the membrane. This equation was proposed elsewhere (ref. 10 in the original submission, ref. 15 in the revised text). We compared the values of the fitted parameters with those obtained by the impedance and chronopotentiometric measurements. The values are consistent within 10-30% error.

Concerning the variation of the parameters: our systematic studies over several recent years showed that the membrane bulk resistance depends on the concentration of the solution. In the revised text we briefly explain this, although we do not go in-depth. Because of this we had to add a number of references to our own works.

  1. The caption of Fig 4 is not clear and does not describe the condition for the experimental curve.

Corrected.

  1. Some paragraphs have only one sentence, such as l132, l150, l227. Should the sentence be combined with the aforementioned paragraph or expanded with more discussion?

Corrected.

  1. The table 1 and 2 are aligned to the left of the page. The layout of the table should be checked.

Corrected.

Round 2

Reviewer 1 Report

The authors have modified the manuscript both in terms of adding new results and correcting/explaining some previously unclear points. The presented research results and improved paper generally will be interesting for the readers. In my opinion, the paper has enough scientific level to be accepted for publication. 

Author Response

Thank you.

Reviewer 3 Report

The authors responded to my concerns to a large extent.  Several more suggestions for authors to improve. 

  1. the abstract does not include the rationale for why curve fitting works for this work.
  2. In the method part, references on electropolymerization procedure should be included. 
  3. in line 136, the use of colon should be checked.
  4. In figure 1, the theoretical result should be provided to prove that results are in line with the ideal Nernstian response. 
  5. What is the differences for the result of three electrodes in figure 1. Are they error bar for the same experiment measurement?
  6. Why should Cottrellian term be adapted? What is the innovation to include such a term? If it is a significant improvement to the fitting equation, it should be emphasized in the conclusion or abstract. 

Author Response

Thank you. We agree with suggestions 1, 3, 4 and 5. We made the respective changes in the revised manuscript, see below. However, we do not agree with suggestions 2 and 6.

  1. the abstract does not include the rationale for why curve fitting works for this work.

Thank you, it is stated now: “ … using the charge curve fitting instead of using a calibration plot …”

  1. In the method part, references on electropolymerization procedure should be included. 

The respective reference (Ref. 7) is present. It is stated in the text: ”The polyethylenedioxythiophene (PEDOT) layer … was formed by galvanostatic electropolymerization … as described elsewhere [7].” Please, see lines 105-108.

  1. in line 136, the use of colon should be checked.

Thank you, colon deleted.

  1. In figure 1, the theoretical result should be provided to prove that results are in line with the ideal Nernstian response. 

Thank you, the theoretical result (29.07 mV/logaCa) is now provided. Please, see line 144.

  1. What is the differences for the result of three electrodes in figure 1. Are they error bar for the same experiment measurement?

Thank you, the detailed explanation provided. Please, see lines 140-144.

  1. Why should Cottrellian term be adapted? What is the innovation to include such a term? If it is a significant improvement to the fitting equation, it should be emphasized in the conclusion or abstract. 

The Cottrellian term allows considering the transportation limitations when a current flows across an ISE membrane. We have demonstrated this in our earlier work (Ref. 15). Therefore, no innovation here. In the revised text we have explained it further, please, see lines 201-204.